# The Prescribing and Education of Naloxone in a Large Academic Medical Center

**DOI:** 10.3390/pharmacy8010031

**Published:** 2020-03-05

**Authors:** Sabrina Miller, Lauren Williams, Amy N. Thompson

**Affiliations:** 1College of Pharmacy, The University of Michigan, Ann Arbor, MI 48109, USA; sabrinmil@med.umich.edu (S.M.); laurewil@med.umich.edu (L.W.); 2Michigan Medicine Pharmacy Innovations and Partnerships, Ann Arbor, MI 48109, USA

**Keywords:** naloxone, education, indication, pharmacists, opioid epidemic

## Abstract

The opioid epidemic has led to increased needs for opioid reversal agents which require education and counseling for proper use. The purpose of this study was to evaluate outpatient naloxone prescribing and education practices at an academic medical center to understand the current state and inform quality improvement measures. This retrospective chart review study included 439 patients that were at least 18 years old and received an outpatient prescription for naloxone between 1 July 2017 and 30 June 2018. Descriptive and demographic data were collected. The primary endpoint was whether an indication for naloxone and education on administration were documented when naloxone was initially prescribed to patients. Overall, 39% of naloxone prescriptions did not have an indication for prescribing listed in the medical record. Of those with a documented indication, concomitant benzodiazepines and history of overdose or substance abuse were most common (22% and 14%). The average morphine milligram equivalents were 165. Additionally, 69% of dispenses did not have documentation that the patient or a caregiver received education regarding the use and administration of naloxone. These findings suggest that patients are receiving naloxone for appropriate indications. Documentation of medication education is needed to ensure it is occurring and that patients are informed.

## 1. Introduction

The Department of Health and Human Services (HHS) declared the opioid epidemic a public health emergency in 2017 [1]. That year, reports from the Centers for Disease Control (CDC) indicated that of the 70,237 overdose deaths, 47,621 deaths (67.8%) were attributed to opioids and 17,144 deaths (24.4%) were attributed to prescription opioids [2]. In addition, the 2017 National Survey on Drug Use and Health estimated that 11.4 million people had misused opioids and approximately 2.1 million people aged 12 years and older had an opioid use disorder [3]. As a result, health systems, insurance companies, and legislative bodies are implementing new guidelines, policies, and regulations to encourage safe opioid prescribing practices. In addition, the Department of HHS has identified five strategies to mitigate dependence, addiction, and death from opioids, one of which includes better availability of overdose reversal drugs such as naloxone [4].

To increase access to naloxone, forty-two states and the District of Columbia have standing orders permitting the distribution of naloxone to patients without a prescription from their health care providers [5]. In addition, various organizations have published recommendations on who should receive a prescription for naloxone. The CDC recommends that naloxone be prescribed for patients taking > 50 morphine milligram equivalents (MME) per day, stating the risk for overdose at least doubles if receiving opioids equaling MME > 50 as compared to < 20 MME [6]. The Surgeon General recommends that naloxone be available for a wide variety of people including patients, family members, health care providers, and community members that may come into contact with someone at risk for an opioid overdose [7]. Known risk factors for an opioid overdose include co-prescribing of benzodiazepines or central nervous system (CNS) depressants, history of a previous overdose, alcohol and illicit drug use, and high-dose (≥50 MME), long-acting, or chronic use of opioids (≥3 months) [8]. 

There are many counselling points that individuals should be aware of regarding the proper administration of naloxone for its use to be successful in reversing an opioid overdose. Individuals must first recognize the signs of an overdose, then administer naloxone correctly, place the patient in the recovery position, call 911, and administer a second dose if necessary. A recent systematic review assessing acceptability, looking for providers’ awareness and willingness to prescribe naloxone, and feasibility, looking at programmatic implementation, education (how providers and patients were educated), of naloxone prescribing in the primary care setting. A total of 17 articles were included in this review and noted that provider willingness to prescribe naloxone has increased over the years the lack of knowledge regarding prescribing and education of naloxone use for patients were significant barriers for providers [9]. The purpose of our study was to evaluate outpatient prescriptions for naloxone as well as education practices over the course of a year at an academic medical center to complete a current state analysis and inform quality improvement measures.

## 2. Materials and Methods 

A retrospective chart review was performed on patients that were at least 18 years old and received an outpatient prescription for naloxone between 1 July 2017 and 30 June 2018. Demographic and descriptive data including age, gender, history of overdose, medical marijuana use, tobacco use, alcohol and illicit drug use were collected for each patient.

The primary endpoint for this study was to identify the percentage of prescriptions with documented indication, based on information within the prescription or provider note from date of prescribing, and percentage of prescriptions with documented education of naloxone provided at time of prescribing within our institution. If education was deferred to a pharmacist, a search for a pharmacist encounter and note of the education was completed. Information was also collected on active benzodiazepine and opioid prescriptions at the time of naloxone prescribing. Name, strength, formulation, and directions for the opioid were recorded and each individual prescriptions’ morphine milligram equivalents (MME) were determined using the CDC Opioid Guidelines Calculator [10]. Total MME for all active narcotic prescriptions was then calculated. In addition, the formulation of naloxone prescribed and any documented naloxone use were captured. Descriptive statistics were used.

## 3. Results

This study reviewed 439 patient charts. In total, there were 500 naloxone prescriptions reviewed, due to patients being prescribed naloxone more than once. On average, patients included in the study were 52 years of age and 57% were female. There was a history of overdose in 19% and history of drug abuse in 28% of the patients, Table 1. The top four pain diagnoses for which opioids were prescribed were chronic, cancer, back, and acute pain.

Patients had an average of 1.5 prescriptions for opioids, with an average total MME of 165, median MME of 90, and range from 0 to 2400. It should be noted that approximately 10% of our patients had a MME of > 400, and more than half, 56%, were on therapy for cancer related pain. In 39 instances, there were no active opioid prescriptions when naloxone was prescribed, with the majority of patients having current or history of drug abuse (n = 23). Patients had a concomitant prescription for benzodiazepines in 54% of the time (n = 271). Additional prescription data are presented in Table 2.

Of the 439 charts reviewed, 39% had no indication documented within the medical record for naloxone therapy. Of the instances where an indication was documented, the most common was concomitant benzodiazepine use followed by a history of overdose or substance abuse. Most often, education was not documented (69%). It is unknown whether this is due to inconsistent documentation practices or whether counseling did not occur. In seven instances, counselling was referred to a pharmacist, yet no patients had an encounter documenting that the education was completed, as shown in Table 3.

The majority of the prescriptions were written for naloxone nasal spray (97%). There were 49 patients who received two prescriptions for naloxone and five patients who received greater than or equal to three prescriptions each. Reasons for why patients may have received more than one naloxone prescription include insurance coverage, product expiration, or refilling the prescription due to use. There was a total of 18 patients with documented use of naloxone after having received the prescription. Administration was completed by first responders, family, friends, and strangers. This number may be underestimated due to patients receiving medical care at outside institutions, not seeking medical attention, or administering to an individual other than the prescribed patient. 

## 4. Discussion

The results of our study show that of the prescriptions with a documented indication, the most frequent factor that led to naloxone prescribing was concomitant benzodiazepine use, followed by a history of overdose or substance abuse. In addition, many patients, 67%, had opioid prescriptions meeting the CDC thresh hold of 50 MME. This implies that prescribers are aware of risk factors for an opioid overdose and are prescribing naloxone appropriately. Data were not captured on patients with similar risk factors that were not prescribed naloxone, thus conclusions on the consistency of prescribing practices cannot be drawn. However, previous studies have concluded that knowledge around who should be prescribed naloxone, the fear of offending patients, and limited time during patient interactions are barriers to prescribing. [11] Creating clinical decision support tools that automatically alert the provider to prescribe naloxone when indicated is a potential avenue to address these barriers. 

Various indications for prescribing naloxone can be easily identified in the electronic medical record to create alerts for providers such as concomitant benzodiazepine prescriptions, the total MME for active opioid prescriptions, social history details, and the patient’s problem or diagnoses list. Alert fatigue and inconsistent documentation in the electronic medical record are both potential barriers. After the completion of this study, including naloxone in order sets with opioid prescriptions when a benzodiazepine was active was implemented. However, anecdotally, there have been several instances where neither the prescriber nor the patient was aware that a naloxone prescription was sent to the pharmacy. The lack of discussion, let alone education, can lead to confusion at the pharmacy and missed opportunities for education. Moving forward, if clinical support tools are created, prompting a discussion rather than automatically sending a prescription may improve patient and provider experience. Including the patient’s specific risk factors in future clinical support tools, along with links to guidelines and counselling points may also improve providers’ level of comfort with prescribing naloxone.

Appropriate education of how naloxone works, as well as when and how to utilize naloxone, is extremely important. Pharmacists are uniquely positioned to ensure proper naloxone education in both community and hospital settings. In some states, pharmacists’ scope of practice also includes the ability to prescribe naloxone for patients that they believe to be at risk [12]. According to the National Association of Chain Drug Stores (NACDS), greater than 91% of people living in America live within 5 miles of a pharmacy. This ease of access, along with legal support in the form of standing orders for dispensing of naloxone products, makes pharmacists valuable resources in the health and safety of at-risk patients.

On the inpatient side, pharmacists can serve as leaders in developing strategies to promote education and proper prescribing. For example, pharmacists at The Johns Hopkins Hospital created patient education materials and provided education sessions with patients [13]. Additionally, the establishment of a pharmacist-led naloxone clinic in North Carolina led to 84% of at-risk patients being educated on naloxone and 69% of those patients filling their naloxone prescription, demonstrating the positive impact pharmacy presence can have in primary care efforts against opioid overdoses [11].

Our study demonstrated a lack of education documentation, with 69% of patients prescribed naloxone not having education or counseling documented in their medical record. There are two possibilities that this fact establishes—either patients are not receiving the necessary education about naloxone and its administration, or they are receiving education, but it is not adequately captured in their chart. With a medication that has the potential to save lives, whether individuals have been informed about its proper use should not be ambiguous. One example noted in our chart reviewed revealed documentation in the record of a patient stated that their friend had to use the intramuscular (IM) formulation to save them. However, the friend did not have instructions and was unsure of how to properly administer the IM. No adverse sequelae were documented. However, this incident further highlights the need for education.

In order to ensure education is given to each patient prescribed naloxone, it is conceivable that a standardized note template located within a specific area in the electronic medical record (EMR) could be implemented. This standardized template should capture the importance of having naloxone in case of overdose, how it works, and how it should be administered. Additionally, instructions for seeking medical care after use should be included. Other tools that could be implemented include medication guides or pamphlets. Attempts at standardized documentation of education should be tailored to individual institutions. Some may find that implementing a workflow in which their outpatient pharmacy utilizes a standard template within the medical record may be an option, though this presents challenges as some patients utilize outside pharmacies. Another possibility may be that clinical pharmacists embedded within clinics and inpatient units be responsible for the education and documentation of any new naloxone prescription on their service.

In conclusion, our study demonstrated that naloxone prescribing practices were appropriate, based off of chart review, but lacked appropriate documentation of indication as well as patient education.

## Figures and Tables

**Table 1 pharmacy-08-00031-t001:** Patient Characteristics.

Patient Characteristics	Frequency (n = 439) N (%)	%
Gender		
Female	250 (56.9%)	56.9
Male	189 (43.1%)	43.1
History of overdose	81 (18.5%)	18.5
History of drug abuse	124 (28.2%	28.2
Medical marijuana use	54 (12.3%)	12.3
Alcohol abuse		
Current	6 (1.4%)	1.4
Former	57 (13%)	13
Tobacco use		
Current	120 (27.3%	27.3
Former	183 (41.7%)	41.7
Diagnosis		
Chronic pain	139 (31.7%)	31.7
Cancer pain	119 (27.1%)	27.1
Back pain	101 (23%)	23
Acute pain	31 (7.1%)	7.1
Drug abuse	28 (6.3%)	6.3
Other ^1^	21 (4.8%)	4.8

^1^ Other pain diagnoses included sickle cell anemia, neuropathy, and arthritis.

**Table 2 pharmacy-08-00031-t002:** Total Morphine Milligram Equivalent (MME) Distribution for Active Opioid Prescriptions.

Total MME	Frequency (n = 500) N (%)	%
0	39 (7.8%)	7.8
1–49	126 (25.2%)	25.2
50–99	134 (26.8%)	26.8
100–149	47 (9.4%)	9.4
150–199	33 (6.6%)	6.6
200–249	28 (5.6%)	5.6
250–299	13 (2.6%)	2.6
300–349	16 (3.2%)	3.2
350–399	13 (2.6%)	2.6
400–2400	51 (10.2%)	10.2

**Table 3 pharmacy-08-00031-t003:** Documented Indication and Education for Naloxone Prescription.

	Frequency (n = 439) N (%)	%
Indication		
No indication listed	171 (39%)	39
Concomitant benzodiazepine	96 (21.9%)	21.9
History of overdose or substance abuse	63 (14.3%)	14.3
Opioid use	61 (13.9%)	13.9
MME > 50	18 (4.1%)	4.1
Family request	4 0.9%)	0.9
Miscellaneous	26 (5.9%)	5.9
Education		
Education documented	128 (29.2%)	29.2
Referred to pharmacist	7 (1.6%)	1.6
No education documented	304 (69.2%)	69.2

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
