# Peer review of "The Prescribing and Education of Naloxone in a Large Academic Medical Center"

_pharmacy, 2020, doi:10.3390/pharmacy8010031_

Round 1
Reviewer 1 Report
Thank you for the opportunity to review your study. Very well described and executed retrospective study of a very topical subject matter. A few comments to help clarify the study are below:
Intro - In general, very well written. I would really clarify in this section that this isn't a review of if all patients who should get naloxone are having it prescribed but rather that it is a review of Rxs written for naloxone. I feel that you're trying to do this but it's not as clear in the intro. It clarifies this greatly in your discussion at the end. I would even consider bringing one of the studies you mention in the discussion into the intro to show that the data on if people are being prescribed naloxone appropriately is already published (and to give the reader a little context as to what this data is). I feel this will help the reader know what to expect within your results a bit better. Results - Line 76 - this is basically a repeat of line 69 explaining why there are 500 rxs without 500 patients I would consider either in results or in the discussion commenting a little on why your average MME was so high when well over 50% of patients had an MME < 100. This is also a good time to just double check those with the MME > 400 and maybe comment on these patients generally since there were 10% here (were they cancer patients? palliative care?). It's not an issue with study design and doesn't impact your outcome, it just really makes your average look very high due to this group. I would even consider showing the median as well to help the reader understand how that 10% between 400-2400 MME skews that data. For Table 3 or somewhere within results/discussion, I would consider just listing the % of patients who do have an MME>50 even though this isn't documented. In a way, just showing the reader that, while only 4.1% documented this as the reason for prescribing naloxone, that actually X% meet this criteria, it just isn't documented. Line 93-95. Just to clarify, is this statement that all who received more than 1 confirmed due to coverage, expiration or refilling due to use or is it possible that they just were prescribed it a 2nd time (possibly unintentionally which is an over use of health care)? If it is confirmed, I may just clarify that. Discussion Line 101 - As mentioned in results, it may be worth telling us how many people met criteria for the CDC MME threshold or had concomitant benzodiazepines and compare that to those who listed that as the reason for prescribing naloxone. I feel this just helps clarify that these scrips were likely appropriate, just not documented appropriately. You mention and discuss this, I just think providing the values would give the full picture. Line 158 - I'm not sure I'm comfortable stating that prescribing practices were appropriate based on the data presented since 39% of those were without a documented reason for prescribing. I'm not saying that they weren't but with the current data provided of 39% lacking an indication, I'm not sure this statement is valid. I would either re-phrase and focus on the education portion or if you provide data showing that these were likely appropriate, just lacked appropriate documentation of indications, then this statement is valid with some adjustments in phrasing.Again, thanks for the opportunity to review and I hope the comments are helpful toward your final product. Strong work!
Author Response
Thank you so much for all of your feedback, please see the attachment.

Reviewer 2 Report
Introduction:
line 46: define high dose and chronic use here line 52: please expand here, what did the systematic review include and specific details of the findings? (necessary to include so that readers know where your study fits in existing literature
Material Methods
line 60: I highly suggest revising to be "% of prescriptions with indication specified" and "% of dispenses with documentation of education". While I see descriptive statistics are used only, the primary endpoint still needs to be stated as a specific outcome which can be measured. line 65: citation needed for CDC calculator
Results
Table 1: I recommend format adjustment for ease of reading. Can the sub-portions (for example male and female gender) in column 1 be indented? Also, I would just use "n (X%)" in one column for ease of reading. Line 78, add range of MME Line 79 "likely due to...." can you report the exact number of those patients without concurrent opioid who did have documentation of prior abuse? Line 83 -- what was your criteria for "documented indication"? I do not think it has been stated. Please add here or in methods above. I would be careful in saying "did not have indication" as opposed to "no indication documented in medical record" as these have two different meanings and implications. Line 93 - 95: were the ones with documented use the same ones with repeat prescriptions? Please specify. To your knowledge, those who had administration of naloxone and were followed up at your facility, was mortality rate 0%?
Discussion
Line 100: You cannot say most common reason was concomitant benzo, since >1/3rd did not have documented indication - right? Line 101-104, I agree. It seems wording in manuscript at times indicated naloxone is prescribed without indication but then here you state it is appropriate. Please read through and edit wording to be consistent. Perhaps highlighting the lack of documented indication while highlighting that most received > 50 MME. Also, commenting on prescriptions for those using opioid without RX -- is that appropriate? line 143 - add "one example noted in our chart review..." or similar to delineate you are providing an example actual scenario
Overall
I suggest reading manuscript and editing for "active voice" throughout. As stated above, modifying tables for visual appeal and being consistent/clear with indication/appropriateness of naloxone prescribing will enhance the article.
Author Response
Thank you so very much for your comments. Please see the attachment.
